# Coupling of Temporal-Check-All-That-Apply and Nose-Space Analysis to Investigate the In Vivo Flavor Perception of Extra Virgin Olive Oil and Carriers’ Impact

**DOI:** 10.3390/foods14132343

**Published:** 2025-07-01

**Authors:** Danny Cliceri, Iuliia Khomenko, Franco Biasioli, Flavia Gasperi, Eugenio Aprea

**Affiliations:** 1Center Agriculture Food Environment, University of Trento, Via E. Mach, 38098 San Michele all’Adige, TN, Italy; flavia.gasperi@unitn.it; 2Edmund Mach Foundation—Research and Innovation Centre, Via E. Mach, 38098 San Michele all’Adige, TN, Italy; iuliia.khomenko@fmach.it (I.K.); franco.biasioli@fmach.it (F.B.)

**Keywords:** EVOO, food carrier, TCATA, nose-space analysis, PTR-ToF-MS, multisensory interaction

## Abstract

The perceived quality of extra virgin olive oil (EVOO) arises from the multisensory integration of multimodal stimuli, primarily driven by non-volatile and volatile organic compounds (VOCs). Given that EVOO is frequently consumed in combination with other foods, cross-modal interactions, encompassing both internal and external elements, play a crucial role in shaping its sensory perception. A more realistic representation of EVOO perception can be achieved by considering these cross-modal effects and their temporal dynamics. This study employed dynamic sensory and instrumental techniques to investigate the product-related mechanisms that influence EVOO flavor perception. Ten trained panelists (mean age = 41.5 years; 50% female) evaluated two EVOO samples under two consumption conditions: alone and accompanied by a solid carrier (bread or chickpeas). Temporal Check-All-That-Apply (TCATA) and nose-space analysis using Proton-Transfer-Reaction Time-of-Flight Mass Spectrometry (PTR-ToF-MS) were conducted simultaneously. Sensory descriptors and mass spectral peaks were analyzed through temporal curve indices (Area Under the Curve, Maximum Citation/Concentration, Time to Maximum), which were then used to construct multi-dimensional sensory and VOC release maps. Findings revealed that the composition and texture of the food carriers had a greater influence on temporal flavor perception than the variability in VOCs released by the different EVOO samples. These results underscore the importance of considering cross-modal sensory interactions when predicting EVOO flavor perception. The carriers modulated both the perception and VOC release, with effects dependent on their specific composition and texture. This methodological approach enabled a deeper understanding of the dynamic relationship between VOC release and EVOO sensory experience.

## 1. Introduction

### 1.1. The Sensory Profile of Olive Oil

Olive oil, a cornerstone of the Mediterranean diet [1], is increasingly consumed globally due to its recognized health benefits and distinct flavor profile [2]. The sensory characteristics of olive oil are critical in shaping consumer preferences and consumption patterns [3,4]. Its complex flavor is the result of the interplay between non-volatile and volatile compounds. Key non-volatile components include phenolic compounds, which elicit gustatory sensations such as bitterness and trigeminal responses like pungency and astringency. Conversely, volatile organic compounds (VOCs) are responsible for the aroma, perceived through both orthonasal and retronasal pathways, and contribute notes described as grassy, tomato leaf, or artichoke [5]. The majority of these VOCs are synthesized via the lipoxygenase (LOX) pathway, an enzymatic cascade that transforms polyunsaturated fatty acids into various five- and six-carbon volatile molecules [6,7], including hexanal and (E)-2-hexenal, which are associated with green aromas, and 1-penten-3-one, linked to leafy notes [8].

### 1.2. Determinants of Volatile Organic Compound Release and Perception

The release and subsequent perception of VOCs in olive oil are influenced by a combination of product-inherent and consumer-related factors. Among the product-related variables, botanical origin, agricultural techniques, and processing technologies are established determinants of the final VOC profile [9]. Furthermore, the food matrix with which olive oil is consumed, often referred to as a carrier food, can significantly modulate the perceived flavor intensity [10]. For instance, proteins have been shown to bind with VOCs, potentially suppressing aroma perception, whereas carbohydrates and phenolic compounds may facilitate aroma release [11]. On the other hand, person-related factors, such as oral temperature, salivary composition and flow rate, oral processing behaviors, and respiration rate, exhibit considerable inter-individual variability and can substantially impact the release and perception of aroma compounds [12,13].

### 1.3. The Complexity of Flavor Perception

The relationship between the release of VOCs and the perception of flavor is not always direct or linear. While some investigations have identified a positive correlation [14], others have documented an inverse relationship [15,16]. These discrepancies may be attributable to the influence of cross-modal interactions, such as those occurring between texture and flavor [17,18] or between taste and flavor [19,20]. Moreover, the perception of olive oil is a temporal phenomenon, frequently characterized by prolonged sensations arising from polyphenols, which are responsible for attributes like pungency, bitterness, and astringency [21]. As olive oil is typically consumed with other foods, these cross-modal effects extend beyond the oil itself to interactions between the oil and the accompanying food matrix. Consequently, a comprehensive understanding of olive oil perception necessitates a dynamic, time-resolved approach that can capture the evolution of these cross-modal effects.

### 1.4. Methodological Approaches for Dynamic Sensory and Instrumental Analysis

To investigate the temporal dynamics of food perception, dynamic sensory methodologies are employed to capture the complete sensory experience and elucidate time-dependent interactions [18,22]. One such technique is Temporal Check-All-That-Apply (TCATA) [23], which has been utilized to study the temporal sensory perception of various food products [24]. In contrast to methods like Time Intensity [25] or Temporal Dominance of Sensations (TDS) [26], TCATA enables the concurrent tracking of multiple sensory attributes, thereby providing a more comprehensive product description [24,27]. This method has been successfully applied to olive oil, proving effective in discriminating between products over time and in identifying individual differences in sensitivity to polyphenol-related sensations [28].

From an instrumental standpoint, while traditional techniques such as Gas Chromatography (GC), High-Performance Liquid Chromatography (HPLC), and Nuclear Magnetic Resonance (NMR) analysis offer detailed compositional information, they are unsuitable for monitoring the rapid dynamics of flavor release during consumption [29,30,31]. In contrast, direct-injection mass spectrometric techniques provide the necessary speed and sensitivity for real-time tracking of flavor compounds during eating [32]. Among these, Proton-Transfer-Reaction Time-of-Flight Mass Spectrometry (PTR-ToF-MS) has been extensively used for tracing aroma and flavor release [33,34] and has been adopted for quality characterization of olive oil due to its high throughput [35]. This technique facilitates rapid, non-invasive headspace analysis, delivering full mass spectra with high mass resolution and short response times without sample preparation.

### 1.5. An Integrated Approach for In Vivo Flavor Analysis

The coupling of PTR-ToF-MS with nose-space analysis permits the real-time, non-invasive monitoring of VOCs released in the nasal cavity via the retronasal route during consumption [36,37]. Recent research has increasingly focused on the simultaneous application of dynamic sensory evaluation and nose-space analysis, with preliminary studies indicating the promise of this integrated methodology [38,39]. The present study aimed to leverage this combined approach to investigate the mechanisms that underpin the flavor perception of extra virgin olive oil (EVOO). Specifically, TCATA was integrated with PTR-ToF-MS nose-space analysis to concurrently measure sensory descriptors and VOC release in vivo and in real time. To account for the influence of food matrices, this methodology was applied to EVOO samples consumed with different solid carriers, reflecting typical consumption contexts.

## 2. Materials and Methods

### 2.1. Subjects

Ten participants (five males, five females; mean age = 41.5 years, SD = 11.7) were recruited from a prior study on EVOO that employed comparable experimental protocols [28]. Participants were selected to fall within the age range of 20–50 years, a span known to represent full olfactory functioning, and gender balance was ensured. All participants had previously been trained in describing EVOO using the TCATA method [23]. All procedures involving human participants were conducted in full compliance with the ethical standards set forth by the relevant national research ethics committee and in accordance with the principles of the Declaration of Helsinki and its subsequent revisions. Participants were thoroughly informed about the study procedures and subsequently provided written informed consent. They were also made aware that participation was voluntary and that they could withdraw at any point without penalty. The procedures to collect consent for subject participation and experimental activities of the present research have been reviewed and approved by the Ethics Committee of the University of Trento (Protocol No. 2023–22 ESA, approval date 21 December 2023).

### 2.2. Samples

#### 2.2.1. EVOO and Carrier Foods

The selection of two Italian EVOOs from the 2017/2018 production season was guided by a preliminary focus group (n = 5). This group’s purpose was to isolate two monovarietal samples derived from differing cultivars that nevertheless exhibited constrained variability in critical sensory characteristics (e.g., bitterness and pungency) for the purpose of simulating real-world variations. Sample O1 was a monovarietal oil derived from the Olivastra Saggianese cultivar (origin: Tuscany, Italy), while Sample O2 was a monovarietal oil from the Grignano cultivar (origin: Veneto, Italy). The key volatile compounds of the samples, quantified through the procedure reported in [7], are reported in Table 1.

Sufficient quantities of each oil from a single production batch were procured and stored at –20 °C until used for sensory characterization. During the experimental phase, oils were kept in their original containers at 4 °C in the absence of light, and were served at room temperature for evaluations.

Two solid carrier foods commonly consumed with EVOO were also selected for their differing nutritional compositions: white bread (Pan Bauletto Bianco, Mulino Bianco, Barilla, Italy; Nutritional value for 100 g: Fat 3.9 g, Carbohydrates 48.9 g, Fibers 4.5 g, Proteins 8.5 g, Salt 1.25 g) and chickpeas (Ceci giganti, Montello, D’Amico, Italy; Nutritional value for 100 g: Fat 2 g, Carbohydrates 17 g, Fibers 5.9 g, Proteins 8.1 g, Salt 1 g). Sufficient quantities for the full study were purchased and stored at room temperature. Bread was portioned into crumb squares (1 cm × 1 cm × 0.5 cm; 0.6 g ± 0.01 g), left uncovered for 20 min to allow ethanol evaporation from preservation, and subsequently stored in sealed containers. Chickpeas were drained, rinsed with tap water, and stored in closed containers (average weight: 1.9 g ± 0.05 g). The dimensions and weights of the carriers were selected to require a comparable number of masticatory cycles (~10 bites) before swallowing. Samples were individually retrieved from storage containers immediately prior to presentation to prevent moisture loss.

#### 2.2.2. Sample Presentation

Nine samples were distributed across three separate trays. Tray 1 always began with a warm-up sample, followed by a randomized presentation of O1 and O2 across participants. In Tray 2, a plain bread square was presented first, followed by randomized presentations of O1B and O2B (EVOO applied to bread). Tray 3 began with a plain chickpea, followed by O1C and O2C (EVOO applied to chickpeas) in randomized order.

EVOO samples (1.5 mL each) were presented in squeezable transfer pipettes, each labeled with a unique three-digit code. Each sample was served with a plastic teaspoon: empty in Tray 1, containing bread in Tray 2, and containing chickpeas in Tray 3. Trays were presented monadically in a fixed order across all participants.

### 2.3. Experimental Procedure

Participants attended four sessions over a one-month period. The first session (described in Section 2.3.1) was dedicated to re-training the panel on relevant sensory attributes and familiarizing them with the study procedures. The subsequent sessions (Sessions 2–4; described in Section 2.3.2) were devoted to the collection of sensory and nose-space data.

#### 2.3.1. Training

Panelists were re-trained on seven sensory attributes identified in a previous phase of this study [28] (see Table 2). These included polyphenol-related sensations commonly found in EVOO (bitterness, astringency, pungency), key descriptors of the fruity sensory profile of EVOO (grass, artichoke, tomato, ripened olive) [40], and attributes relevant to the selected carriers. Training sessions introduced the updated evaluation procedures integrating TCATA with nose-space analysis, and participants engaged in a simulated test session using EVOO samples.

Training took place in individual sensory booths under white light conditions. Sensory data were recorded using Fizz software (version 2.51, Biosystèmes, Couternon, France). Panel performance was evaluated during a preceding phase of this study, following the procedure reported in [28]. The performance was assessed in terms of replication index for each attribute using the tempR package [41]. An arbitrary replication index threshold of 0.6 was employed to define acceptable replicability for each assessor, considering average replication index scores in [23]. At the end of the training phase, the average agreement index amongst the assessors was 0.69.

#### 2.3.2. EVOO Samples Evaluations

##### Tasting Procedure

Panelists were instructed to begin each evaluation session after a relaxation period of at least 30 s to stabilize their breathing rhythm. Following this, they were trained to dispense the EVOO sample from the pipette onto the corresponding spoon, introduce the contents into the mouth, and then press the “Start” button displayed on the screen.

Depending on the type of sample, different oral procedures were followed. For pure EVOO samples (tray 1), panelists were instructed to hold the oil in the mouth and move it around for 5 s without swallowing [21]. For samples containing bread or chickpeas (trays 2 and 3), assessors were asked to chew at a controlled pace of one bite per second for 10 s. At the end of the designated time, a message on the screen prompted the participant to swallow the sample. The total duration of each evaluation was 150 s.

Between sample evaluations, participants performed a standardized cleansing procedure: rinsing the mouth with water for 20 s, eating a plain cracker for 20 s, and then rinsing again for 20 s. Between the evaluations of different trays, participants were given a 5 min break and were allowed to leave their booths if desired. Each sample was evaluated in triplicate, with each repetition completed in a separate, individual session.

##### TCATA Evaluation

During the tasting procedure, panelists conducted a TCATA evaluation. According to this methodology, participants continuously selected the sensory descriptors that were appropriate to describe their perception of the sample and deselected those that no longer applied. The list of attributes, displayed on a screen, varied according to the type of sample being assessed:EVOO only: Bitter, Astringent, Pungent, Grass, Artichoke, Tomato, Ripened olive;Bread only: Bread, Sweet;Chickpea only: Chickpea, Salty;EVOO with bread: Bitter, Astringent, Pungent, Grass, Artichoke, Tomato, Ripened olive, Bread, Sweet;EVOO with chickpea: Bitter, Astringent, Pungent, Grass, Artichoke, Tomato, Ripened olive, Chickpea, Salty.

Evaluations were performed individually, in dedicated sensory booths illuminated with white light. Each booth was connected to a PTR-MS apparatus for simultaneous nose-space analysis. TCATA data were collected using Fizz software (version 2.51, Biosystèmes, Couternon, France).

##### Nose-Space Analysis

Simultaneously with the TCATA evaluation, VOCs release in the nasal cavity was monitored using a commercial Proton Transfer Reaction Time-of-Flight Mass Spectrometer (PTR–ToF–MS 8000; Ionicon Analytik GmbH, Innsbruck, Austria). Sampling was performed using an ergonomic glass nosepiece connected to the instrument via a PEEK tube. Once the nosepiece was comfortably positioned in the nostrils, participants were instructed to breathe normally through the nose (with the mouth closed) while following the tasting procedure described above.

Expired air was drawn through a heated PEEK sampling tube (maintained at 110 °C), with the final 40 cm left at ambient temperature for participant comfort. The sampling inlet flow rate was set to 240 cm^3^ STP/min to ensure optimal time resolution. Ionization settings included a drift voltage of 628 V, a drift temperature of 110 °C, a drift pressure of 2.80 mbar, and the use of an active ion funnel, yielding an E/N ratio of 130 Townsend (1 Td = 10^−17^ cm^2^ V^−1^ s^−1^). Data acquisition was configured to collect one full mass spectrum per second [37] across a mass range of *m*/*z* 21 to *m*/*z* 205.

### 2.4. Data Analysis

#### 2.4.1. Temporal Curves and Parameter Selection

Citation occurrences from the TCATA evaluation were computed for each sensory attribute, product, and time point (1 s intervals) by summing responses across all participants and replications. These occurrences were expressed as citation percentages—defined as the proportion of the maximum possible citations for each attribute and product at a specific second (e.g., 100% indicates that all participants cited the attribute at that time point across all replications, while 0% indicates no citations). Temporal perception of each attribute for a given product was represented through citation percentage curves over the 150 s evaluation period [23].

From each temporal curve, the following descriptive parameters were extracted: (i) Area under the curve (AUC)—the cumulative sum of citations over time, representing overall intensity across the evaluation period [28,42]; (ii) Maximum citation percentage (C_max)—the peak value of citation percentage during the evaluation period; (iii) Time to maximum citation (T_max)—the elapsed time in seconds from the start of the evaluation to the point at which C_max was reached.

For each of these parameters, a Principal Component Analysis was conducted based on the values of flavor attributes for all evaluated samples, with the aim of exploring patterns of sensory perception.

#### 2.4.2. Nose-Space Curves and Parameter Extraction and Selection

Preprocessing of nose-space data, including dead time correction, internal calibration of mass spectral data, and peak extraction, was performed in accordance with established procedures [43,44]. The raw mass spectrometry dataset initially comprised 246 ion peaks ranging from 21 to *m*/*z* 205. Application of a peak-like feature selection algorithm [37] yielded a reduced set of 58 peaks. Further elimination of signals attributable to impurities, water clusters, and isotopologues narrowed the dataset to 22 meaningful ion peaks. Tentative identification of VOCs was carried out based on their exact masses and known fragmentation patterns of compounds typically found in EVOO [45].

For each of the 22 selected mass peaks, the baseline signal, calculated as the average intensity during the 30 s preceding sample intake, was subtracted. Three analytical parameters were then extracted for each peak: (i) Area under the curve (AUC)—computed from the moment the sample entered the mouth and extending for 120 s; (ii) Maximum intensity (I_max)—the peak ion count during the evaluation period; (iii) Time to maximum intensity (T_max)—the time in seconds required to reach I_max after sample intake [36,46].

For each of these parameters, a mixed-effects ANOVA model was applied to test the influence of sample and replication on each ion peak (R package lmer ver. 1.1-23; model: CONC ~ SAMPLE + (1|JUDGE); CONC ~ REPLICATION + (1|JUDGE)). Only mass peaks showing a statistically significant sample effect and no significant replication effect (α < 0.05) were retained. Subsequently, a Principal Component Analysis was conducted for each parameter based on the selected peaks to explore variation in the VOC profiles across samples. All statistical analyses were performed using R software ver. 4.0.1.

## 3. Results

### 3.1. Temporal Curves

Temporal curves enabled the assessment of differences between EVOO samples and the influence of pairing with accompanying matrices on their temporal sensory profiles. All selected sensory attributes significantly discriminated among samples, particularly during the “attack” phase (i.e., the first 40 s of evaluation) (Figure 1). This discriminative capacity was primarily observed between EVOO samples consumed alone and those consumed with a carrier matrix, whereas no significant differences were found between the two EVOO samples when evaluated alone.

Overall, the presence of a carrier had a more pronounced impact on attributes related to polyphenols than on “fruity” meta-attributes. Specifically, the addition of bread attenuated the perception of polyphenol-related sensations (i.e., pungent, bitter, astringent) during the initial phase of consumption. Bread also modulated specific flavor perceptions, diminishing the Artichoke note while enhancing the Grass note. A similar, though less extensive, trend was observed with chickpeas, primarily affecting the Pungent attribute.

The extent of modulation by the carrier varied depending on the EVOO sample. Bread had a greater impact on sample O1 than on O2, particularly concerning polyphenol-related descriptors (Bitter, Astringent) and the “fruity” Grass note. Conversely, chickpeas had a greater influence on polyphenol-related sensations in O1 and on “fruity” descriptors (Ripened olive, Tomato) in O2.

### 3.2. Nose-Space Curves

Following ANOVA-based validation (i.e., significant sample effect and no replication effect), seven mass peaks were retained for each of the three parameters: AUC, I_max, and T_max (Table 3). Further details regarding the statistical validation are provided in the Appendix A. The annotation VOCs listed in Table 3 was performed by cross-referencing experimental data with both the scientific literature, including our prior studies using GC-MS and PTR-MS [7,45,47], and an internal library of olive oil headspace standards. According to the Metabolomics Standards Initiative, this approach achieves a Level 2 confidence (putatively annotated compounds based on spectral matching to databases/the literature). It is noteworthy that the employed nose-space analysis preferentially detects the most abundant and volatile species. The high signal intensity and clear spectral profiles of these dominant compounds enhance the reliability of their identification by significantly reducing the probability of interference from co-eluting or minor analytes, thus corroborating the robustness of the final assignments.

### 3.3. Relationship Between Flavor Attributes and VOCs

The multivariate analysis of sensory and instrumental data enabled investigation of the relationship between flavor perception and VOC release.

With respect to the AUC (Figure 2B), sample grouping was primarily driven by the presence or absence of a carrier matrix. EVOO samples paired with chickpeas showed a lower overall flavor perception compared to those paired with bread or consumed alone. A secondary trend indicated that O2 generally elicited a stronger flavor perception than O1. Pairing with bread enhanced certain descriptors (e.g., Grass, Ripened olive) while suppressing others (e.g., Artichoke, Tomato), relative to EVOO consumed alone.

Regarding the AUC of VOC release (Figure 2A), sample differentiation was primarily based on the EVOO type, with O2 exhibiting a higher release than O1. To a lesser extent, pairing with bread also led to greater VOC release, followed by chickpeas, and lastly EVOO alone.

Specific VOCs responded differently to the carrier and EVOO type. Mass peak ms85.0642 (tentatively identified as 1-Penten-3-one) and ms83.0813 (t.i. Hexanal + (E)-2-Hexen-1-ol + (Z)-3-Hexenol) varied with the carrier matrix, while ms197.1395 (t.i. unknown), ms117.0897 (t.i. Hexanoic acid), and ms99.0799 (t.i. (E)-2-Hexenal) differed based on EVOO type.

For the maximum citation/intensity parameter (Figure 3), the grouping of samples followed a similar pattern to the AUC analysis, albeit with less pronounced differences between carrier types. Maximum intensities of individual VOCs also varied as a function of both carrier and sample: ms119.0923 (t.i. 3-Phenylpropanol), ms85.0642 (t.i. 1-Penten-3-one), and ms83.0813 were influenced by the carrier, whereas ms117.0897, ms99.0799, and ms57.0407 (t.i. Acrolein) differed by EVOO type.

In contrast, time to maximum citation/intensity produced a distinct pattern of sample grouping. Nose-space and sensory temporal data showed aligned configurations (Figure 4), with longer times to maximum citation/intensity observed for samples paired with carriers, and shorter times for those consumed alone. Flavor attributes such as Artichoke, Tomato, and Ripened olive exhibited delayed peaks when EVOO was combined with chickpeas.

The time to maximum VOC intensity also varied by both carrier and EVOO type. For example, ms85.0642 (t.i. 1-Penten-3-one), ms87.0798 (t.i. Pentanal), and ms99.0799 (t.i. (E)-2-Hexenal) responded to the presence of carriers, whereas ms205.1983 (t.i. Sesquiterpenes), ms197.1395 (t.i. unknown), and ms143.1013 (t.i. (Z)-3-Hexenyl acetate) were influenced by EVOO type.

## 4. Discussion

### 4.1. EVOO Perception and Carrier Impact

The dynamic sensory method did not reveal significant differences between the two investigated EVOOs when evaluated in isolation. In contrast, distinct perceptual differences emerged when the EVOOs were assessed in association with solid carriers. Moreover, the type of solid carrier employed had a differential effect on flavor perception. Specifically, the use of bread attenuated polyphenol-related sensations and selectively modulated flavor perception, suppressing attributes such as Artichoke while enhancing others like Grass. A similar, albeit more limited, trend was observed with chickpeas, which primarily influenced the perception of Pungency. These findings align with previous research indicating that the mouth coating effect of a food can decrease taste intensity, and that food matrix consistency strongly impacts oral burn sensations, with solid food matrices generally suppressing oral burn intensity [48,49]. The divergent sensory outcomes are likely attributable to the distinct physical properties of the solid carriers, which govern the release kinetics of the EVOO’s volatile and non-volatile components. The porous structure of bread, characterized by an extensive surface area, enhances the release of volatile compounds while simultaneously enabling higher absorption and a more gradual release of the oil [16,50]. This sustained release mechanism is proposed to underlie the observed modulation of flavor perception. Conversely, the denser and more viscous texture of chickpeas is hypothesized to induce a more immediate release of the EVOO, leading to a different sensory impact [15,51]. The interplay between the carrier’s physical matrix (e.g., porosity, surface area, texture) and the oil’s components is therefore a critical factor in the dynamic perception of flavor. These findings align with previous research on the co-consumption of food and carriers with respect to VOC release, which has shown that carrier addition can increase in vivo aroma release while diminishing overall aroma perception [39,52]. The carrier’s impact likely transcends physical structure, involving compositional factors. The bread matrix’s macromolecules (i.e., starch and gluten proteins) can entrap volatile compounds and bind polyphenols [53,54]. This protein-polyphenol complexation would reduce the oral accessibility of phenols, thereby attenuating bitterness and pungency. The distinct macromolecular profile of chickpeas may confer a differential binding affinity for specific phenolic classes, accounting for the more targeted suppression of pungency. 

Polyphenol-related sensations have been shown to exert a suppressive effect on flavor perception, both in aqueous solutions [19,20,45] and in EVOO matrices [28]. This suppression may result from cognitive mechanisms, wherein pungency becomes more salient than other flavor attributes. Consequently, the attenuation of polyphenol-related sensations by the carrier may alleviate this suppression, thereby enhancing flavor perception both qualitatively and quantitatively. Additionally, congruence between taste and olfactory cues may amplify specific sensory attributes. In the present study, a congruent enhancement was observed for the Artichoke attribute, linked to bitter taste and astringency.

Despite the overall similarity between the two EVOO samples, the magnitude of the carrier effect varied depending on the sample. The EVOO with a higher concentration of polyphenol-related attributes exhibited a more pronounced response to the carrier, leading to greater modulation, either suppressive or enhancing, of other flavor perceptions. Therefore, to achieve a more ecologically valid assessment of consumer experience, evaluating EVOO in conjunction with a carrier is advisable, particularly for samples with intense and persistent polyphenol-related sensations.

### 4.2. VOCs Release and Carrier Impact

Variation in VOCs among the samples had a comparatively lesser impact on flavor perception than variation in carrier composition. Nevertheless, the trend in VOC variation aligned with flavor perception patterns. The EVOO sample characterized by a higher VOC release (O2) consistently elicited greater citation frequencies of flavor descriptors compared to the alternative sample.

The type of carrier significantly influenced VOC release, although this effect was intertwined with oral processing. Chewed samples exhibited enhanced release of specific compounds, such as those corresponding to mass signals ms85.0642 (tentatively identified as 1-Penten-3-one) and ms83.0813 (tentatively identified as a mixture of Hexenal, (E)-2-Hexen-1-ol, and (Z)-3-Hexen-1-ol), particularly in the case of bread. These compounds, derived from lipoxygenase activity on olive oil polyunsaturated fatty acids, are associated with green sensory attributes such as Leaf and Tomato notes [8,55,56]. Consistent with the prior literature, the increased release of these VOCs was found to positively correlate with the temporal perception of the Grassy attribute in in vivo measurements.

The elevated release of specific VOCs, in conjunction with the reduced suppressive effect of polyphenol-related sensations, highlights the significant influence of bread as a carrier on the temporal sensory profile of EVOO. Identifying VOCs that are preferentially released in the presence of particular carriers may serve as a predictive tool for evaluating consumer experience. For EVOOs rich in such compounds, sensory differences between evaluations with and without carriers may be particularly pronounced.

### 4.3. Temporal Sensory and Instrumental Assessment of EVOO Flavor

This study employed a combined methodological approach that enabled the simultaneous assessment of temporal flavor perception and the instrumental measurement of VOCs reaching the nasal cavity via retro-nasal pathways. This dual approach provided valuable insights into the temporal dynamics of multiple sensory descriptors while ensuring higher ecological validity compared to conventional in vivo temporal sensory assessments [57].

Findings indicated that while VOC measurements are critical to flavor perception, they alone are insufficient to fully explain human sensory responses. Notably, differences in VOC composition between the two EVOO samples did not translate into discernible differences in temporal flavor perception, although similar trends were observed. In contrast, the choice of carrier resulted in more pronounced differences in temporal sensory profiles than in the variation between EVOO samples.

As with the evaluations without carriers, EVOO consumed with bread elicited higher citation rates for flavor descriptors than when consumed with chickpeas. This may be due to the structural differences between the carriers; bread, with its larger surface area, may facilitate greater VOC release. Furthermore, the carbohydrate–to–protein ratio in the carriers may influence flavor perception, given that proteins can bind to volatile compounds, suppressing their perception, while carbohydrates may enhance it [11]. Additionally, the slower release of EVOO from the bread matrix may reduce exposure to polyphenol-related sensations, thereby lessening flavor suppression. Oral processing, however, appeared to have a limited and non-directional impact on flavor perception, as no consistent effects were observed between the two processing types (chewing and tongue movement vs. tongue movement only).

### 4.4. Limitations and Future Directions

This study utilized two EVOOs characterized by relatively similar sensory profiles. Although this selection facilitated the simulation of real-world differences across samples, it could concurrently restrict the broader generalizability of the results. EVOOs with more distinct sensory characteristics might yield different results, with VOC variation potentially exerting a greater impact than the carrier. In such cases, instrumental and perceptual assessments may exhibit greater consistency. Future research should consider incorporating a broader range of EVOO samples, particularly those differing in polyphenol content and composition, as well as various carriers with diverse textures within the same category. Such an expansion could elucidate additional effects of VOCs and carriers on temporal perception.

A limitation of using real-food carriers like bread and chickpeas is that the effects of physical texture and chemical composition on perception were confounded. To disentangle these variables, future studies could employ model systems. This would involve developing gels with identical, rheologically confirmed textures but distinct compositions (e.g., protein-only, starch-only, and an inert control). Assessing the olive oil in these standardized carriers would allow for the quantification of each macromolecule’s specific contribution to the sensory profile.

The TCATA methodology did not detect significant differences between the EVOOs when evaluated without carriers but proved effective in revealing perceptual differences when carriers were present. This limitation may stem from the nature of the differences identified during sample selection, which were primarily quantitative. TCATA is specifically designed to elucidate the temporal dynamics of sensory perception. It effectively identifies which sensory attributes are perceived and charts their appearance and disappearance throughout the evaluation period. However, a notable limitation of TCATA is its inability to directly quantify the intensity of these perceived attributes. While it provides robust data on the presence or absence of various sensory characteristics, it does not measure their perceived strength. Consequently, TCATA may exhibit reduced sensitivity to variations in attribute intensity across different samples, in contrast to its strong capacity for discerning qualitative differences [23]. Consequently, this characteristic may have hindered the effective investigation of associations with increased VOC concentrations, thereby attenuating potential correlations. Alternative methods, such as Time Intensity, offer more precise temporal profiling but are constrained by the limited number of descriptors that can be evaluated due to lengthy test durations [58]. A promising alternative is the Discrete Time Intensity method [59], which accommodates a broader range of descriptors. Nonetheless, TCATA remains a valuable tool for preliminary descriptor screening prior to applying more intensive methodologies.

## 5. Conclusions

This study proposed an integrated in vivo approach for the simultaneous evaluation of EVOO through temporal flavor perception and the release of VOCs via the retro-nasal pathway. This methodology enabled the concurrent monitoring of the temporal evolution of multiple sensory descriptors alongside the quantification of VOC concentrations in the nasal cavity.

In the two EVOO samples examined, characterized by comparable sensory profiles, variation in the type of solid carrier (in terms of composition and texture) exerted a more pronounced influence on temporal flavor perception than the differences in VOC release. In contrast, the effect of oral processing appeared to be relatively minor. Notably, the use of bread as a carrier led to an enhanced release of specific VOCs (e.g., 1-Penten-3-one) and concurrently reduced the suppressive influence of polyphenol-related sensations on flavor perception, thereby exerting a substantial impact on the temporal sensory profile of EVOO.

The Identification of volatile compounds that are more efficiently released in the presence of specific carriers may serve as a valuable predictor of consumer experience. This is particularly relevant for EVOOs rich in such compounds, for which the sensory differences between evaluations with and without carriers may be especially pronounced.

## Figures and Tables

**Figure 1 foods-14-02343-f001:**
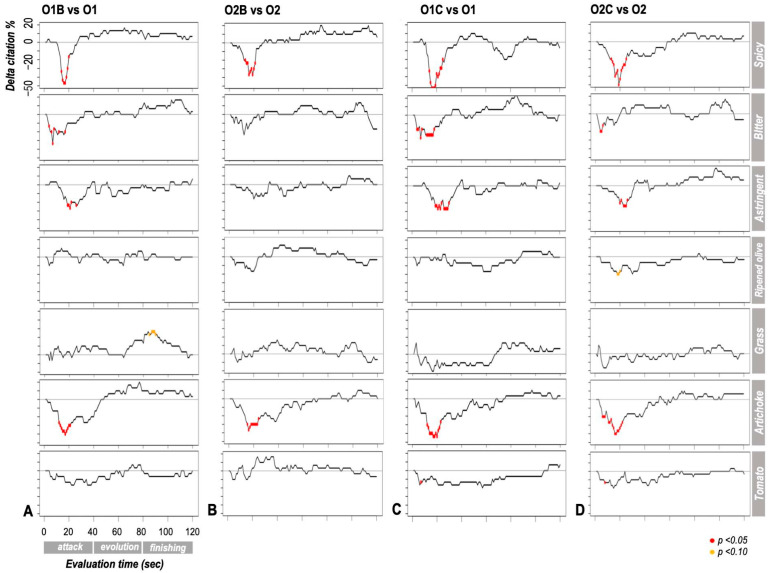
Difference in citation percentage (Δ citation %) for each attribute during the temporal sensory evaluation (Temporal Check–All–That–Apply) between (**A**) sample O1 alone vs. with bread (O1B), (**B**) sample O2 alone vs. with bread (O2B), (**C**) sample O1 alone vs. with chickpeas (O1C), and (**D**) sample O2 alone vs. with chickpeas (O2C). At each time point, negative values indicate a higher citation percentage for the sample consumed on its own. Points marked with red dots represent statistically significant differences (*p* < 0.05), while those with yellow dots suggest a marginal significance or tendency (*p* < 0.10).

**Figure 2 foods-14-02343-f002:**
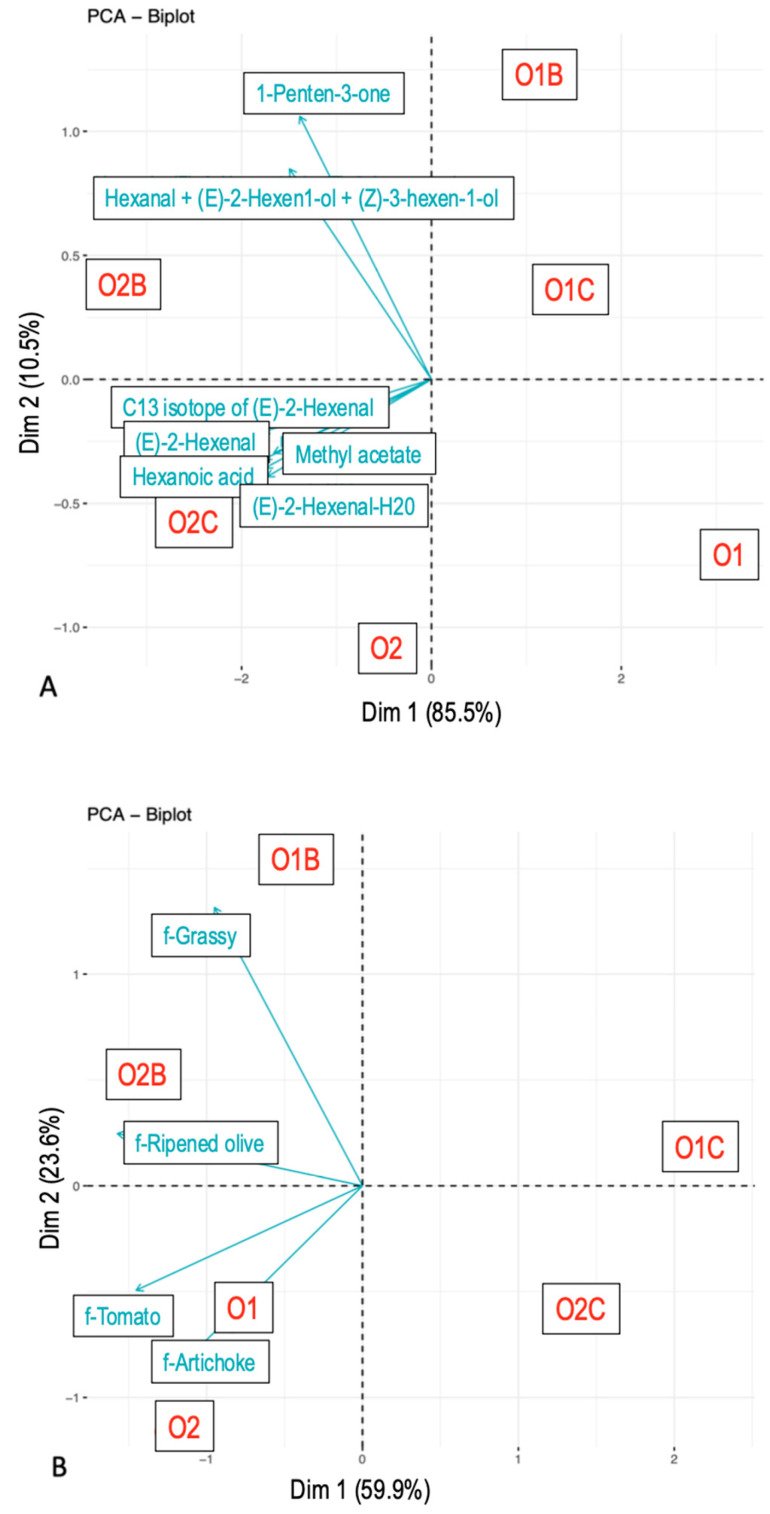
Principal component analysis. Area under the curve values from nose-space analysis curves (**A**) and Temporal Check-All-That-Apply curves (**B**). Compounds were tentatively identified. The samples included O1—olive oil sample 1, O2—olive oil sample 2, O1B—O1 with bread, O2B—O2 with bread, O1C—O1 with chickpeas, and O2C—O2 with chickpeas.

**Figure 3 foods-14-02343-f003:**
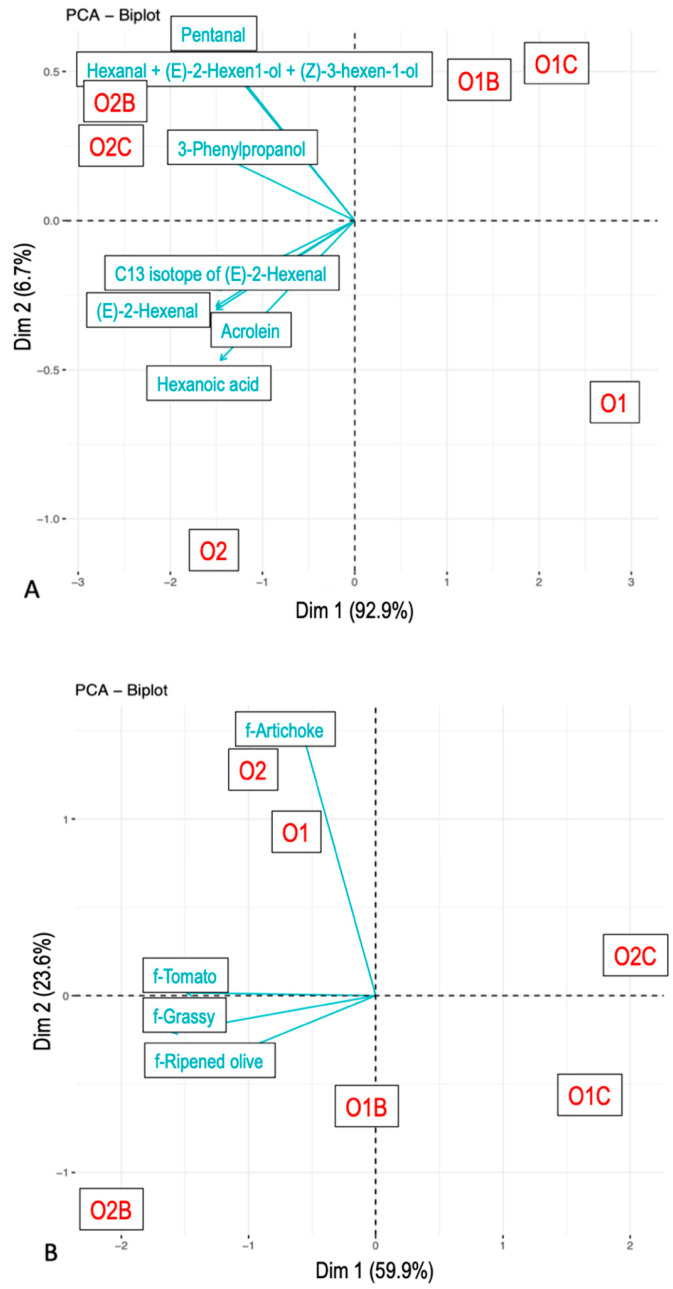
Principal component analysis. Maximum concentration from nose-space analysis curves (**A**) and citation rate from Temporal Check-All-That-Apply curves (**B**). Compounds were tentatively identified. The samples included O1—olive oil sample 1, O2—olive oil sample 2, O1B—O1 with bread, O2B—O2 with bread, O1C—O1 with chickpeas, and O2C—O2 with chickpeas.

**Figure 4 foods-14-02343-f004:**
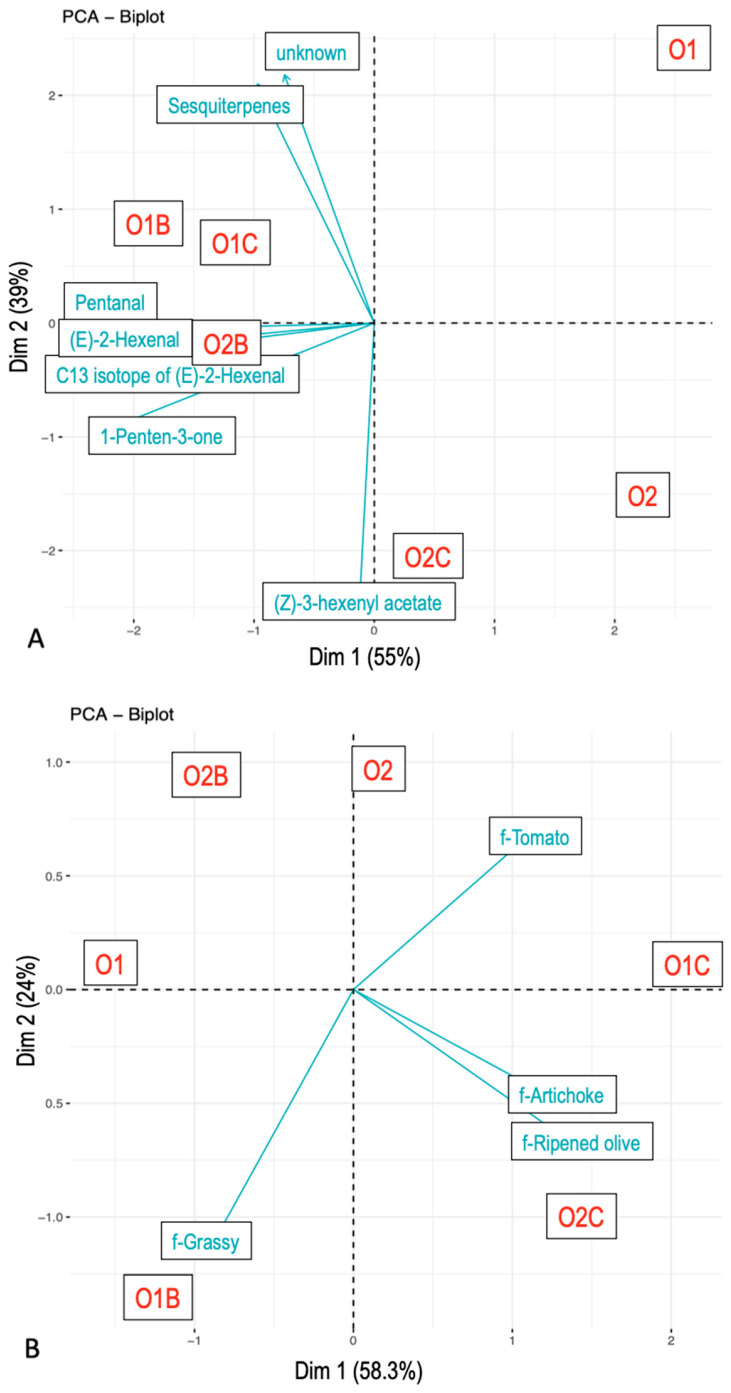
Principal component analysis. Time of the maximum concentration from nose-space analysis curves (**A**) and citation rate from Temporal Check-All-That-Apply curves (**B**). Compounds were tentatively identified. The samples included O1—olive oil sample 1, O2—olive oil sample 2, O1B—O1 with bread, O2B—O2 with bread, O1C—O1 with chickpeas, and O2C—O2 with chickpeas.

**Table 1 foods-14-02343-t001:** Key volatile compounds in olive oils (μg/Kg).

Compound	Samples	
	O1	O2
Ethyl acetate	862	139.3
1-Pentanol	303.2	25.3
1-penten-3-ol	564	103.4
1-penten-3-one	948.9	107.3
3-penten-2-one	158	52.4
(Z)-3-Hexenyl acetate	275.4	187.5
Pentanal	268.8	22.5
Hexanal	83.3	58.2
Hexyl acetate	1583.9	125.1
(E)-2-Hexenal	477.8	193.2
(E)-2-Hexen-1-ol	2385	6.3
(Z)-3-Hexen-1-ol	841.5	83.8
(E,E)-2,4-Hexadienal	1968.6	54
Limonene	2492.1	706.8
Linalool	64.4	110.6
α-Farnesene	294	1.3
Heptanal	491.5	7
(E,E)-2,4-Heptadienal	-	15
1-Octen-3-ol	56.5	8.4
(E)-2-Octenal	3.4	13
(E)-2-Nonenal	453.6	33
Nonanal	85.4	1
Decanal	62.4	0.3
(E,E)-2,4-Decadienal	4.1	31.7
Dodecanal	90	7.9

**Table 2 foods-14-02343-t002:** Name, description, and standard for sensory descriptors used in Temporal Check-All-That-Apply evaluations with EVOO samples.

Attribute	Description	Standard
Bitter	Taste associated with bitter foods such as coffee, quinine	caffeine 1.5 g/L in water
Pungent	Tingling sensation on the tongue and throat, associated with a feeling of warmth	capsaicin 0.36 mg/L in water
Astringent	Sensation of dryness and roughness perceived on the tongue and in the oral cavity	potassium aluminum sulfate 0.8 g/L in dezionized water
Grass	Flavor associated with freshly cut grass	cis-3-hesene-1-ol 0.07 g/kg in seed oil
Artichoke	Flavor associated with fresh artichoke	40 g of artichoke stem in 250 g of seed oil prepared 24 h before evaluation
Tomato	Flavor associated with tomato and tomato leaf	33 g of cherry tomato in 250 g of seed oil prepared 24 h before evaluation
Ripened olive	Flavor associated with mashed olives	1 g of dried olive skin and pulp

**Table 3 foods-14-02343-t003:** Protonated masses or mass fragments retained after statistical validation, respectively, for each parameter describing the temporal curves from nose-space analysis.

Parameter	Mass	Compound (Tentatively Identified)
AUC	ms117.0897	Hexanoic acid
	ms100.0825	C13-isotope of (E)-2-Hexenal
	ms99.0799	(E)-2-Hexenal
	ms85.0642	1-Penten-3-one
	ms83.0813	Hexenal + (E)-2-Hexen-1-ol + (Z)-3-hexen-1-ol
	ms81.0690	(E)-2-Hexenal -H20
	ms75.0444	Methyl acetate
I max	ms119.0923	3-Phenylpropanol
	ms117.0897	Hexanoic acid
	ms100.0825	C13-isotope of (E)-2-Hexenal
	ms99.0799	(E)-2-Hexenal
	ms87.0798	Pentanal
	ms83.0813	Hexanal + (E)-2-Hexen-1-ol + (Z)-3-hexen-1-ol
	ms57.0407	Acrolein
T max	ms205.1983	Sesquiterpenes
	ms197.1395	Unknown
	ms143.1013	(Z)-3-Hexenyl acetate
	ms100.0825	C13-isotope of (E)-2-Hexenal)
	ms99.0799	(E)-2-Hexenal
	ms87.0798	Pentanal
	ms85.0642	1-Penten-3-one

AUC—Area under the curve; I max—Maximum concentration of the compound; T max—time to reach the I max.

## Data Availability

The raw data supporting the conclusions of this article will be made available by the authors on request.

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
