# Peer review of "Coupling of Temporal-Check-All-That-Apply and Nose-Space Analysis to Investigate the In Vivo Flavor Perception of Extra Virgin Olive Oil and Carriers’ Impact"

_foods, 2025, doi:10.3390/foods14132343_

Round 1
Reviewer 1 Report
Comments and Suggestions for Authors
The manuscript presents a well-executed and novel study that combines dynamic sensory analysis (TCATA) with in vivo VOC measurements (PTR-ToF-MS) to assess the influence of different food carriers on the flavor perception of extra virgin olive oil (EVOO). The experimental design is sound, the data are clearly presented, and the conclusions are well supported. However, a few clarifications and minor revisions are necessary to improve the manuscript's clarity and scientific rigor.
- While the authors mention that the two EVOO samples were selected to have limited variability in sensory characteristics, it would be useful to include a more detailed description (e.g., phenolic content, free acidity, or volatile profile) to contextualize the comparability of these oils. This would help interpret why perceptual differences were not significant when oils were evaluated alone.
- The discussion appropriately notes that TCATA may not detect intensity differences effectively. However, it would benefit from citing any pilot comparison or quantitative data (if available) supporting this specific limitation in the current dataset.
- Table summarizing key physical/nutritional properties (e.g., fat, protein, water content, texture metrics) of the bread and chickpeas would enhance understanding of how these factors might modulate flavor release and perception. This would also align better with the discussion in Sections 4.1 and 4.2.
- The manuscript mentions a mixed-effects ANOVA but does not provide details on random vs. fixed effects or on the software used for analysis. Adding a short description of the model structure and software (R version + packages used) would increase reproducibility.
- Panelist performance is briefly discussed (Section 2.3.1), but more quantitative information on repeatability and discrimination would strengthen this section. For example, consider reporting mean panel performance indices (e.g., city block distances or other metrics from tempR).
- In Table 2, it is unclear whether VOCs were confirmed using standards or inferred solely based on mass and fragmentation. Please clarify the confidence level of compound identification (e.g., level 2 or 3 according to Metabolomics Standards Initiative).
- Figures 2–4: Consider enlarging axis labels and legends for better readability. Ensure compound identifications (e.g., "ms85.0642") are explained or cross-referenced in figure captions for clarity to readers unfamiliar with mass spectrometry.
Author Response
We thank the reviewer for the positive feedback and pertinent comments. We have addressed all suggestions, detailing our responses in blue within this rebuttal document and highlighting the corresponding modifications in blue within the revised manuscript. We trust that our revisions adequately address the points you have raised.
The manuscript presents a well-executed and novel study that combines dynamic sensory analysis (TCATA) with in vivo VOC measurements (PTR-ToF-MS) to assess the influence of different food carriers on the flavor perception of extra virgin olive oil (EVOO). The experimental design is sound, the data are clearly presented, and the conclusions are well supported. However, a few clarifications and minor revisions are necessary to improve the manuscript's clarity and scientific rigor.
While the authors mention that the two EVOO samples were selected to have limited variability in sensory characteristics, it would be useful to include a more detailed description (e.g., phenolic content, free acidity, or volatile profile) to contextualize the comparability of these oils. This would help interpret why perceptual differences were not significant when oils were evaluated alone.
We agree that a detailed description of the oils' characteristics is crucial for contextualizing the findings. Our selection criterion for the two EVOO samples was primarily sensory similarity/dissimilarity, as our study specifically aimed to investigate the relationship between instrumental and sensory analyses. This approach was chosen given the demonstrated lack of direct correlation between certain physicochemical parameters and sensory perception, as highlighted in our manuscript. To address the reviewer's comment, we have now included a table in the revised manuscript (Table 1, page 4) that summarizes the main physicochemical characteristics of the EVOO samples. We believe this addition provides the necessary context for understanding the inherent variability and comparability of the oils.
The discussion appropriately notes that TCATA may not detect intensity differences effectively. However, it would benefit from citing any pilot comparison or quantitative data (if available) supporting this specific limitation in the current dataset.
We acknowledge that our study did not include quantitative intensity scales alongside the TCATA methodology. Therefore, a direct comparison of TCATA's performance in detecting intensity differences against a more traditional intensity-based method cannot be directly tested using the data collected within this specific dataset. However, we want to emphasize that the observation about TCATA's potential limitation in effectively detecting subtle intensity differences is a recognized characteristic of the method within the broader scientific literature. We have now clarified this point in the revised manuscript's discussion section (Lines 502-503), referencing relevant studies that support this inherent limitation of the TCATA approach when compared to direct intensity scaling methods.
Table summarizing key physical/nutritional properties (e.g., fat, protein, water content, texture metrics) of the bread and chickpeas would enhance understanding of how these factors might modulate flavor release and perception. This would also align better with the discussion in Sections 4.1 and 4.2.
We thank the reviewer for the suggestion. We've incorporated into the text the samples nutritional properties (Lines 141-143).
The manuscript mentions a mixed-effects ANOVA but does not provide details on random vs. fixed effects or on the software used for analysis. Adding a short description of the model structure and software (R version + packages used) would increase reproducibility.
We thank the reviewer for the suggestion. We have added further details on the statistical analysis to the text (Lines 288-289).
Panelist performance is briefly discussed (Section 2.3.1), but more quantitative information on repeatability and discrimination would strengthen this section. For example, consider reporting mean panel performance indices (e.g., city block distances or other metrics from tempR).
We agree that incorporating more quantitative information on panel performance will strengthen Section 2.3.1. We've clarified that the performance evaluation procedure is consistent with that used in an earlier phase of this study, which is fully detailed in a separate publication [31]. This approach allowed us to minimize redundancy in the methods section of the current manuscript. Additionally, we've included further quantitative details on the repeatability index (Lines 188-193)
In Table 2, it is unclear whether VOCs were confirmed using standards or inferred solely based on mass and fragmentation. Please clarify the confidence level of compound identification (e.g., level 2 or 3 according to Metabolomics Standards Initiative).
We thank the reviewer for the important remark. The identification of VOCs reported in former Table 2 (now Table 3) is based on a combination of available literature data — particularly from our previous work on olive oil aroma using both GC-MS and PTR-MS techniques — and our own extensive reference measurements on olive oil headspace [7, 48, 56]. In this sense, our compound annotation aligns with level 2 of the Metabolomics Standards Initiative (putatively annotated compounds based on spectral similarity and comparison with literature and/or database entries).
It is important to note that nose-space analysis inherently favors the detection of the most abundant and volatile compounds—i.e., those present at sufficient concentrations to produce significant signals. This, in turn, enhances the confidence of identification for the dominant compounds. Given the high signal intensity and clear spectral features of these dominant VOCs, the likelihood of interference from minor or co-eluting compounds is greatly reduced, further supporting the robustness of our assignments.
We have clarified this in the revised manuscript (Lines 325-335).
Figures 2–4: Consider enlarging axis labels and legends for better readability. Ensure compound identifications (e.g., "ms85.0642") are explained or cross-referenced in figure captions for clarity to readers unfamiliar with mass spectrometry.
As requested by the reviewer, we've enlarged the text in Figures 2-4. Because Figures 2-4 list compound names and do not include mass information, we believe their clarity is already guaranteed for readers unfamiliar with mass spectrometry.
Reviewer 2 Report
Comments and Suggestions for Authors
This study establishes a combined approach of sensory evaluation and instrumental analysis for food flavor profiling, providing valuable reference for flavor analysis in complex food systems. The manuscript contains substantial data supporting its conclusions. I suggest addressing the following points:
- Avoid abbreviations in keywords where possible.
- The introduction contains excessive paragraph fragmentation (e.g., lines 97-107). Suggest reorganizing by grouping related content and focusing on: Key research advances, Study significance.
- Provide more details about "Two Italian EVOOs" .
- Lines 307-309: Are there similar studies reporting consistent conclusions?
- Improve resolution of Figures 2-4.
- Line 392: Specify which physical properties of the carrier are being referenced.
Author Response
We thank the reviewer for their positive feedback and pertinent comments. We have addressed all suggestions, detailing our responses in blue within this rebuttal document and highlighting the corresponding modifications in blue within the revised manuscript. We trust that our revisions adequately address the points you have raised.
This study establishes a combined approach of sensory evaluation and instrumental analysis for food flavor profiling, providing valuable reference for flavor analysis in complex food systems. The manuscript contains substantial data supporting its conclusions. I suggest addressing the following points:
Avoid abbreviations in keywords where possible.
While we acknowledge the suggestion to avoid abbreviations in keywords, we opt to retain them in the manuscript. The terms represented by these abbreviations appear very frequently throughout the text. Using the full terms in the keywords would lead to significant redundancy and potentially hinder readability and searchability. Furthermore, we have thoroughly ensured that all acronyms are clearly defined and explained in the captions of all figures and tables, as well as upon their first mention in the main body of the manuscript. We believe this approach balances clarity and conciseness, especially given the repetitive nature of these specific terms in our work.
The introduction contains excessive paragraph fragmentation (e.g., lines 97-107). Suggest reorganizing by grouping related content and focusing on: Key research advances, Study significance.
We have revised the Introduction chapter, specifically addressing the fragmentation identified by the reviewer. We've now reorganized the content into thematic subsections, which we hope will reduce the perception of fragmentation and significantly improve readability.
Provide more details about "Two Italian EVOOs" .
We have revised the manuscript to incorporate a summary of the nutritional characteristics of the EVOO samples (Lines 141-143). Furthermore, we have elucidated the selection criteria in the 'Samples' section (Section 2.2, lines 128-130), as detailed in our response to Reviewer 1.
Lines 307-309: Are there similar studies reporting consistent conclusions?
Our findings align with previous research indicating that the mouthcoating effect of a food can decrease taste intensity, and that food matrix consistency strongly impacts oral burn sensations, with solid food matrices generally suppressing oral burn intensity. We have added this specification in the text (Lines 404-407).
Improve resolution of Figures 2-4.
We've enlarged the text and improved graphic quality in Figures 2-4 as requested by the reviewer.
Line 392: Specify which physical properties of the carrier are being referenced.
We thank the reviewer for the suggestion. In the text, we are referencing properties such as the porosity, surface area, and texture of the carrier, which directly influence its absorption capacity and the subsequent release kinetics of the EVOO's volatile compounds and non-volatile components. For instance, bread's porous structure facilitates a higher absorption and slower, more gradual release of EVOO, leading to different flavor modulation compared to chickpeas, which likely offer a more immediate release due to their denser structure. We will amend the text to explicitly detail these properties and their implications (Lines 409-420).